

# Riverbank filtration for treatment of highly turbid Colombian rivers

Juan Pablo Gutiérrez[1,2], Doris van Halem[1], Luuk Rietveld[1]

[1] Civil Engineering Department, Delft University of Technology, Delft, 2628CN, Netherlands
[2] Cinara Institute, Faculty of Engineering, Cali, 760032, Colombia

*Correspondence to*: Juan Pablo Gutiérrez (J.P.GutierrezMarin@tudelft.nl; juan.p.gutierrez@correounivalle.edu.co)

**Abstract.** The poor water quality of many Colombian surface waters, forces for seeking alternative, sustainable treatment solutions with the ability to manage peak pollution events and to guarantee an uninterrupted provision of safe drinking water to the population. This review assesses the potential of using riverbank filtration (RBF) for the highly turbid waters in Colombia emphasizing on water quality improvement and the influence of clogging by suspended solids. The suspended sediments may
on the one hand be favorable in the improvement of the water quality mainly due to the strengthening of cake filtration and deep bed filtration processes. On the other hand, the formed cake layer must be balanced by scouring in order for an RBF system to be sustainable without loss of hydraulic capacity. In general, RBF seems to be a technology appropriate for use in highly turbid and contaminated surface rivers in Colombia, where improvements due to the removal of turbidity, and pathogens, and to a lesser extent inorganics, organic matter and micro-pollutants are expected. RBF has the potential to mitigate
shock loads thus leading to the prevention of shutdowns of surface water treatment plants. In addition, RBF, as an alternative pre-treatment step, may provide an important reduction of chemicals' consumption, considerably simplifying the operation of the existing treatment processes. However, clogging and self-cleansing issues must be studied deeper in the context of these highly turbid waters, evaluating the potential loss of abstraction capacity yield as well as the development of different redox zones for efficient contaminant removal.

**1 Introduction**

Riverbank filtration (RBF) is a water abstraction technology that consists of production wells that extract water some distance away from a surface water body (Figure 1). As the production wells pump water from the aquifer, surface water flows underground to recharge it, while the subsurface sediments function as a natural filter removing several contaminants, producing a higher quality water than the raw source water has (Schubert, 2003; Sontheimer, 1980; Tyagi et al., 2013). In
addition, the naturally present groundwater contributes to the higher water quality extracted from RBF systems, e.g. through attenuation (Kuehn and Mueller, 2000) and the change of redox conditions (Bourg, 1992; Hiscock and Grischek, 2002).
The well configuration in RBF systems can be either vertical or horizontal that offer different benefits. Vertical wells are commonly used for seeking longer residence or travel times to ensure higher removal efficiencies of more mobile contaminants. Horizontal wells are usually applied for obtaining higher water flows, but which may be unfavorable for the
quality of the water abstracted due to shorter residence times (Hunt et al., 2003; Ray, 2002b).



Many variables influence the performance of RBF systems, including riverbed media composition and hydraulic connectivity of the aquifer (Hubbs et al., 2007; Hunt et al., 2003; Schubert, 2002). In Europe and the United States, RBF has been widely used, because of the favorable hydraulic conditions (Brunke, 1999; Goldschneider et al., 2007; Hubbs et al., 2007; Stuyfzand et al., 2006; Veličković, 2005). In addition, RBF has a demonstrated ability to be an effective water treatment technology used

in contaminated surface waters (Singh et al., 2010; Thakur and Ojha, 2010).

A key water quality parameter determining the performance of RBF systems is the concentration of total suspended solids (*TSS*) contained in the surface water, since long-term changes in the composition and concentration of suspended solids can have potential cumulative effects on clogging of riverbanks and alluvial aquifers. In addition, suspended solids generally act as the primary transport mechanism for emerging organisms and pollutants (Bourg et al., 1989; Miretzky et al., 2005; Stone

and Droppo, 1994; Zhu et al., 2005). Turbidity is one of the parameters used to indirectly describe the concentration of suspended solids (EPA, 1999), conveniently measured due to the high relationship between both parameters (Susfalk et al., 2008; Wu et al., 2014) and the relatively long analysis time of *TSS* compared to turbidity analysis (Susfalk et al., 2008).

RBF has the additional advantage of removing or attenuating certain heavy metals (Bourg and Bertin, 1993; Stuyfzand, 1998), pathogens (Dillon et al., 2002; Schijven et al., 2003; Schmidt et al., 2003; Sprenger et al., 2014; Weiss et al., 2005) and nutrients

(Krause et al., 2013; Ray, 2002b; Schmidt et al., 2003; Wu et al., 2007). In addition, RBF has demonstrated an ability to decrease mutagenic compounds (Schubert, 2003) and to remove certain organic and inorganic micro-pollutants (Bertelkamp et al., 2014; Hamann et al., 2016; Schmidt et al., 2003). However, it has also been found that specific micro-pollutants remain mobile, showing a persistent behavior even after 3.6 years of travel time (Hamann et al., 2016). RBF has also shown the capacity to mitigate shock loads (Mälzer et al., 2003; Schmidt et al., 2003), resulting in a stable abstracted water quality.

Although RBF has shown to be highly effective in the removal of many contaminants, it must mainly be considered as a pre-treatment method, which needs to be combined with a certain post-treatment. (Cady et al., 2013; Dash et al., 2008; Kuehn and Mueller, 2000; Singh et al., 2010).

Surface water bodies are the main sources used for supplying drinking water to the Colombian communities, being approximately 80% of the systems (Ministerio de Desarrollo de Colombia, 1998). However, in the last decades, turbidity and

contamination events in surface waters have become a serious concern in Colombia for guaranteeing safe drinking water (Gutiérrez et al., 2016; Universidad del Valle and UNESCO-IHE, 2008). Fast urbanization, the lack of integration between water management and spatial planning, and inappropriate land use are identified as the main causes for the progressive deterioration of the surface water (IDEAM, 2015; van der Kerk, 2011; Universidad del Valle and UNESCO-IHE, 2008).

The Pacific basins of Colombia e.g., have sediment yields between 1,150 and 1,714 t/km$^2$/year (Restrepo and Kjerfve, 2004),

while the Magdalena River in the Magdalena-Cauca basin, which corresponds to the most populated zone of the country, has the highest sediment yield (560 t/km$^2$/year) of the large rivers of the Caribbean and Atlantic coasts of South America, having similar yields to those found in the larger basins of South-Asian rivers (Restrepo et al., 2009). In addition, significant loads of heavy metals (up to 122 kg/d Hg; 2,600 kg/d Pb; 3,300 kg/d Cd; 490 kg/d Cr) and nutrients (up to 1,138,000 kg/d N and 769,000 kg/d P) have been found in sediments of the Magdalena River (IDEAM, 2011).



Considering the poor water quality of many Colombian surface waters, there is a need for seeking alternative, sustainable treatment solutions with the ability to manage peak pollution events and to guarantee an uninterrupted provision of safe drinking water to the population. RBF has shown to be effective in the removal of many river water pollutants and can therefore also be of interest for drinking water companies, and environmental and public health authorities in Colombia (Hülshoff et al.,

2009; Schijven et al., 2003; Schmidt et al., 2003; Schubert, 2003).

The few reported experiences using RBF in highly turbid and contaminated surface waters led to conducting this review, assessing the potential of using RBF for the highly turbid waters in Colombia emphasizing on water quality improvement and the influence of clogging by suspended solids.

## 2 Water quality improvement

### 2.1 Mechanisms of water quality improvement in RBF systems

RBF removes contaminants by filtration, sorption of pollutants to soil particles, microbial degradation, chemical precipitation, ion exchange, and oxidation/reduction (Schmidt et al., 2003; Schubert, 2003). In the first centimeters of the riverbed a fine sediments' layer is formed, also known as cake layer. The cake layer is called schmutzdecke if a highly active biological layer is involved (Hiscock and Grischek, 2002; Unger and Collins, 2006). A certain degree of clogging in the riverbed is preferred

since it can be favorable for water quality improvement (Ray and Prommer, 2006), due to the augmentation of traveling times, particulate removal and the richness of processes occurring in the schmutzdecke (Hiscock and Grischek, 2002; Schmidt et al., 2003; Unger and Collins, 2006). Jüttner (1995) determined e.g. that the schmutzdecke and upper layers were responsible for most of the elimination of volatile organic carbon, and Dizer et al. (2004) concluded that this layer is extremely efficient in eliminating viruses. A cake layer, mainly composed of organic and/or clay constituents, may also enhance the sorption of

pollutants onto its surface (Li et al., 2003).

The surface water – groundwater interface, corresponding to the hyporheic zone (Figure 1), plays the most important role in degradation of contaminants (Doussan et al., 1997; Grischek and Ray, 2009; Maeng et al., 2008; Smith et al., 2009; Stuyfzand, 2011). The hyporheic zone is characterized by redox gradients, dynamic exchange of oxygen, presence of organic carbon and microorganisms (Doussan et al., 1997; Febria et al., 2012; Findlay and Sobczak, 2000), which enhance electron transfer, ion

exchange, degradation and sorption processes and therefore improve the removal of pollutants (Hiscock and Grischek, 2002; Smith, 2005; Tufenkji et al., 2002). Commonly, microbial activity is high in the early stages of infiltration conveying to the depletion of oxygen in the hyporheic zone, producing anoxic/anaerobic conditions (Doussan et al., 1997; Krause et al., 2013) The flow path between the river and the abstraction well is characterized by lower biological activity and sorption capacity as well as longer retention times and increased mixing (Hiscock and Grischek, 2002; Stuyfzand, 2011). This flow path is therefore

of great importance for the removal of poorly degradable pollutants, which require greater distances to be removed or inactivated. In both the hyporheic zone and the flow path, deep bed filtration mechanisms are important.



During deep bed filtration, the particles in suspension to be removed are considerably smaller than the average size of the pores of the aquifer (Brunke, 1999; Sutherland, 2008; Zamani and Maini, 2009). Therefore, particle separation mainly occurs due to selective straining within the porous media, through sedimentation, interception, inertial forces or Brownian motion (Sutherland, 2008). Pathogens are mainly removed from the water through straining, inactivation and attachment to the soil

grains  (Schijven et al., 2003).

The transformation of nutrients in the subsoil is a function of the river-hyporheic zone exchange rates, residence times, dissolved oxygen and biotic processes (Krause et al., 2013; Smith, 2005). The hyporheic zone may have anoxic/anaerobic conditions due to high levels of microbial activity (Doussan et al., 1997; Krause et al., 2013). If consumption of oxygen exceeds the hydrological oxygen exchange rate, anoxic conditions lead to an oxic-anoxic interface. Therefore, reduced and oxidized

forms of nutrients may coexist under such conditions (Duff and Triska, 2000).

The removal of heavy metals from source water during subsurface passage mainly occurs by sorption, precipitation and ion exchange processes, which depend on the content of inorganic and organic compounds in the aquifer and contact time (Bourg et al., 1989; Hülshoff et al., 2009). Under aerobic conditions, heavy metals removal is mainly attributed to ion exchange processes. The presence of negatively charged surfaces (e.g. clayey and/or organic sediments) and amorphous ferric and

alumina oxides provide exchange sites for binding trace heavy metals (Foster and Charlesworth, 1996; Salomons and Förstner, 1984). In anoxic aquifers, heavy metals are mainly removed by sorption processes (Schmidt and Brauch, 2008). If the conditions are such that sulfide is formed, the immobilization of heavy metals may occur through sulfide precipitation (Bourg et al., 1989; Salomons and Förstner, 1984).

Micro-pollutants occur in most surface waters that run through heavily polluted regions or large industrial and agricultural

areas. The fate of such substances is mainly determined by adsorption mechanisms and biological transformations (Schmidt et al., 2003). Extensive research in Germany has shown that these compounds may be removed to varying degrees, mainly depending on the properties of each compound (Schmidt et al., 2003).

## 2.2  Turbidity removal at RBF sites with highly turbid surface waters

Turbidity removal has been proven to be highly efficient using RBF (Dash et al., 2008, 2010; Ray et al., 2008; Saini et al.,

2013; Schubert, 2001; Thakur and Ojha, 2010; Wang, 2003; Wang et al., 1996, 2001; Weiss et al., 2005). Thakur and Ojha (2010) e.g. studied the variation of turbidity during the extraction of subsurface water for the supply of drinking water to Haridwar. According to these authors, the river channel (from Ganga River in Uttrakhand, India) reached turbidity values up to 2,500 NTU, obtaining turbidity removals between 99 and 99.9% during RBF. In Table 1, more turbidity removal values are presented from RBF sites with highly turbid surface waters.

The RBF system configuration (i.e. vertical, horizontal) does not govern the suspended solids removal efficiency as observed in Table 1, since it is not a function of the travel/contact time. The texture of the streambed, however, influences the media clogging (Hubbs et al., 2007; Stuyfzand et al., 2006), where external clogging (cake layer formation) enhances the removal capacity of fine sediments contained in the water (Veličković, 2005). The removal efficiency of suspended solids is



concentration dependent (Fallah et al., 2012; Thakur and Ojha, 2010); the higher the suspended solids concentration, the faster the cake formation and therefore the higher the turbidity removal capacity. Although, no studies have quantified the role of concentration on entrapment, the critical particle concentration where the porous media gets clogged has been determined to be dependent on the ratio of void size to particle size (Sen and Khilar, 2006). As reported by Sen and Khilar (2006), the critical

particle concentration increased from 0.35% to 9% when the ratio of bead size to particle size was increased from 12 to 40.

## 2.3 Pathogens removal by RBF

Schijven et al. (2003) showed the efficiency of RBF for microbial contaminant removal, which depends on flow path length and residence times; the longer the flow path and the residence time, the higher the removal. Bacterial removal larger than 2.5-log has been reported in RBF systems with most of the removal occurring in the first meter of filtration (Wang, 2003). Cady

et al. (2013) studied an RBF system in Kali River, achieving removals of 2.7-log for total coliforms and 3.4-log for *E. coli* (being 1-log for *E. coli* per 26 m). However, Weiss et al. (2015) found that total coliforms were reduced at two sites by, on average, 5.5 and 6.1-log respectively.

Virus removal up to 5-log was reported by Sprenger et al. (2014), after only 3.8 m of RBF passage (approximately 8 days of residence time), concluding that RBF is a suitable technology for rivers in emerging countries with regards to viruses removal.

Derx et al. (2013) found that flooding events significantly alter the removal efficiency of viruses in RBF systems by increasing the advection and dispersion of the viruses through the aquifer system. The virus concentration in the abstraction wells was found to increase up to eight times due to the decrease in travel times.

Weiss et al. (2005) reported parasite (*Cryptosporidium* and *Giardia*) removal at three RBF facilities, where no parasites were detected in the well waters. Metge et al. (2010) studied the parasite (*Crystoporidium parvum*) removal efficiency in an RBF

system comprised of well-graded, metal oxide rich content sediments, finding that the main immobilization mechanism was sorption to the metal oxide contents (iron and aluminum).

## 2.4 Nutrients removal by RBF

Doussan et al. (1997) studied the behavior of nitrogen as nitrate, nitrite and ammonium in a RBF system fed by the Seine River. They found a complete removal of nitrate and nitrite, while the ammonium concentrations at the RBF site increased in

comparison to the concentration in the river water. Regnery et al. (2015) also found a significant decrease in nitrate concentrations by denitrification. The presence of reducing conditions is commonly found during RBF passage due to the long paths and residence times of the water transported from the river to the RBF abstraction wells. Ammonium concentrations are usually low in surface waters due to the nitrification processes occurring in rivers. However, even low ammonium concentrations can cause an extensive oxygen reduction during infiltration (Doussan et al., 1997). By contrast, Wu et al. (2007)

reported a decrease in ammonium concentrations and an increase in nitrate and nitrite concentrations in an unsaturated RBF passage, associated with oxic conditions leading to nitrification processes. They reported removals of nitrogen over 95% by



nitrification/denitrification under saturated conditions during the monitoring period. The ammonium concentrations in the river water corresponded to a highly polluted river (16.42 mg/L) (Wu et al., 2007).

Phosphorus is generally removed by sorption and precipitation in the form of calcium, iron or aluminum/iron phosphate (Regnery et al., 2015; Schmidt et al., 2003). Phosphorus removal is influenced by the sedimentary structure of the subsoil
(Hendricks and White, 2000). Its sorption is linked to the exchange between the river water and the soil matrix (Hülshoff et al., 2009; Smith, 2005). Leader et al. (2008) assessed the sorption dynamics for different materials, finding sorption ranging from 66 to 97 mg-P/kg for clean sand and about 515 mg-P/kg for iron-coated sand. As stated by Vohla et al. (2007), the amount of phosphate that can be removed during subsurface passage is limited to the number of sorption sites, leading to a sorption capacity decrease over time, and to changes in the physicochemical and oxidation conditions. Regnery et al. (2015) found a
decrease in the phosphate removal efficiency in a RBF system from 80% during start-up to 40% after 6-years.

## 2.5 Heavy metals removal by RBF

RBF has shown to be a suitable technology to remove certain heavy metals (Bordas and Bourg, 2001; Bourg et al., 1989; Bourg and Bertin, 1993; Stuyfzand, 1998), although its ability is site and substance specific. As pointed out by Sontheimer (1980), Schmidt et al. (2003) and Stuyfzand et al. (2006), some RBF systems are able to remove heavy metals, such as
chromium, and metalloids, like arsenic, by approximately 90%. This in accordance with the experiences with the use of similar technologies like sand filtration, also resulting in the removal of heavy metals (Awan et al., 2003; Baig et al., 2003; Schmidt and Stadtwerke, 1977). Schmidt et al. (2003) also found lead and cadmium removals up to 75% at an RBF site located in Germany, abstracting water from the Rhine River. However, Stuyfzand et al. (2006) found that lead and cadmium concentrations in the abstraction wells increased over 300% and 30%, respectively, in a 450 days travel time. Bourg et al.
(1989) also found that cadmium and zinc were remobilized from sediments, although Bourg and Bertin (1993) still reported zinc removal through river bank sediments.

## 2.6 Micro-pollutants removal by RBF

Hamann et al. (2016) studied the fate of 247 micro-pollutant compounds in a RBF system considering a travel time up to 3.6 years, finding complete removal of 14 compounds (2-naphthalene sulfonate, 2,6-NDS, amidotrizoic acid, AMPA, aniline,
bezafibrate, diclofenac, ibuprofen, iohexol, iomeprol, iopromide, ioxitalamic acid, metoprolol and sulfamethoxazol). In addition, some compounds were partially removed (triglyme, iopamidol, 1,3,5-naphthalene trisulfonate, 1,3,6-naphthalene trisulfonate), and only 10 compounds were fully persistent during the subsurface passage in the RBF system (1,4-dioxan, 1,5-naphthalene disulfonate (1,5-NDS), 2-amino-1,5-NDS, 3-amino-1,5-NDS, AOX, carbamazepine, EDTA, MTBE, toluene and triphenylphosphine oxide).
Bertelkamp et al. (2014) assessed the sorption and biodegradation of 14 organic micro-pollutants (acetaminophen, ibuprofen, ketoprofen, gemfibrozil, trimethoprim, caffeine, propranolol, metoprolol, atrazine, carbamazepine, phenytoin, sulfamethoxazole, hydrochlorothiazide and lincomycin) at laboratory scale, finding that most of them (the first eight



compounds listed before) were completely biodegraded. However, compounds such as atrazine and sulfamethoxazole were not removed in a 6-month period. Schmidt et al. (2003) found that sulfamethoxazole was primarily removed (20% removal efficiency) under anaerobic conditions (anaerobic aquifer), while only slightly reduced in the RBF system under aerobic conditions. Drewes et al. (2003) examined the fate of selected pharmaceuticals and personal care products during groundwater

recharge, stating that the stimulants caffeine, diclofenac, ibuprofen, ketoprofen, naproxen, fenoproxen and gemfibrozil, were efficiently removed. However, the antiepilectics carbamazepine and primidone were not removed at all. Organic iodine was only partially removed.

## 3 Clogging and self-cleansing in RBF

### 3.1  Hydraulic conductivity and clogging of the aquifer

RBF systems worldwide have shown a decline in the long-term yield (Caldwell, 2006; Dash et al., 2010; Hubbs, 2006a; Hubbs et al., 2007; Mucha et al., 2006; Schmidt et al., 2003; Schubert, 2006a; Stuyfzand et al., 2006). The production yield of RBF depends on many factors, including the hydraulic conductivity and the degree of contact between river and the phreatic aquifer (Caldwell, 2006). Temperature affects the production yield seasonally due to changes in water viscosity (Caldwell, 2006; Hubbs, 2006a); however, this parameter is not a concern in tropical countries like Colombia where the temperature in surface

water sources remains largely constant throughout the entire year (Lewis Jr., 2008).

Commonly, hydraulic conductivity varies spatially and can, temporally, be dependent on clogging and interface renewal through scouring. The clogging layer leads to a reduction in hydraulic conductivity of the streambed and then affects the hydraulic connectivity between the river and the aquifer. This alters the surface water/groundwater interaction and therefore may influence the abstraction capacity yield (Brunke, 1999; Packman and MacKay, 2003). Nevertheless, the clogging might

be favorable for quality improvement due to longer travel times and greater particulate removal, as discussed before.

Clogging has been identified as the major contributor to the long-term decay of RBF yield (Hubbs et al., 2007), but there is a lack of understanding of the exact factors that affect clogging (Caldwell, 2006; Hubbs et al., 2007; Schubert, 2006a; Stuyfzand et al., 2006). Hubbs et al. (2007) reported a decrease in the specific capacity of the wells up to 67% of its initial level in the first 4-year period of operation. Most of the reduction took place within the first year due to riverbed clogging in the vicinity

of the well. Clogging is time dependent and is a function of bed material (Goldschneider et al., 2007; Rehg et al., 2005), content and composition of suspended load and transported bed load material  (Bouwer, 2002; Holländer et al., 2005), and the shear forces (Hubbs, 2006b; Schubert, 2006b) scouring out the deposited material on the riverbed (Hubbs, 2006a; Mucha et al., 2006). Clogging can be caused by physical, chemical and biological processes, although physical clogging has been found to be the dominant mechanism over the other forms of clogging (Pavelic et al., 2011; Rinck-Pfeiffer et al., 2000).

As water flows from the river and through the aquifer to the RBF system, the larger silt particles plug the pore channels to the aquifer in the riverbed and form a less permeable layer together with smaller particles (Grischek and Ray, 2009; Veličković, 2005). Tropical river conditions (temperature and nutrient loads) may be favorable for biological growth onto the riverbed,



which might lead to biological clogging (Kim et al., 2010; Platzer and Mauch, 1997; Vandevivere et al., 1995). Rinck-Pfeiffer et al. (2000) reported biological clogging by biomass and bacterially produced polysaccharides in a simulated aquifer storage and recovery wells system, related to the high presence of nutrients. Hoffmann and Gunkel (2011) reported severe clogging mainly induced by biological processes in Lake Tegel reaching a depth of at least 10 cm.

As pointed out by Hubbs et al. (2007), medium coarse sand to fine gravel in the riverbed is desirable, so that only little fine sand and silt can penetrate the larger voids in the aquifer, and therefore, a permanent reduction of the hydraulic conductivity of the aquifer may be avoided. However, Sakthivadivel and Einstein (1970) stated if that when the ratio between the bed particle and the suspended particle is larger than 20, clogging of the bed occurs. Also, experiences from the Netherlands have suggested that riverbeds consisting primarily of gravel (up to 25 cm in size) are at a greater risk of clogging than those

consisting primarily of finer grade materials (Stuyfzand et al., 2006). This is due to the fact that the finer particles will be able to penetrate at a greater distance into the gravel riverbed before clogging (Veličković, 2005). Consequently, there is a reduced chance of resuspension or scouring of these particles; the gravel bed acts as a protective shield from flow shear forces, and infiltration rates become permanently impaired (Goldschneider et al., 2007). In sandy and silty riverbeds, the clogging particles cannot penetrate as deeply, and a cake layer will be formed on the riverbed surface (Brunke, 1999; Veličković, 2005). In these

instances, flood waves will more easily be able to resuspend and remove the clogging particles, thereby regenerating bed infiltration rates to some degree. Levy et al. (2011) estimated a recovery of the hydraulic conductivity by a factor of 1.5 (from 31% to 47% compared to the hydraulic conductivity of the media before clogging).

Aquifers hydraulically connected to surface waters are susceptible to long-term accumulation of micro-sized (colloidal) particles (Baveye et al., 1998; Hiscock and Grischek, 2002; Vandevivere et al., 1995), which causes reduction in the hydraulic

conductivity, leading to reduction in production yield capacity. Hoffmann and Gunkel (2011) reported a decrease in the hydraulic conductivity in a bank filtration system of about two orders of magnitude during the winter period. As stated by Hoffmann and Gunkel (2011), the water temperature decrease only accounted to a change in hydraulic conductivity from $4.8 \times 10^{-4}$ to $3.1 \times 10^{-4}$ m/s. Thus, clogging by micro-sized particles (e.g. particulate organic matter), in combination with atmospheric air intrusion, was considered to be the main factor to reduce the hydraulic conductivity. The clogging of the

aquifer also depends on the concentration and type of micro-sized particles (Zamani and Maini, 2009). As stated by Okubo and Matsumoto (1983), the concentration should be below 2 mg/L to sustain a high infiltration capacity during long inundation periods. In addition, Jacobsen et al. (1997) reported that particles < 10 µm are absorbed more strongly at the macropore wall due to their relatively large surface charge, whilst particles > 10 µm are more exposed to hydraulic force.

## 3.2  Interface renewal by scouring

The deposition of sediments carried by river water on the riverbed surface must be balanced by renewing scouring in order for a RBF system to be sustainable. Naturally occurring flow forces may induce sufficient scouring of the riverbed, thereby self-regulating the thickness of the formed cake layer, scouring the bed and restoring its hydraulic conductivity. Scouring is the result of shear stress forces exerted on the riverbed. The extent of scouring is determined by the magnitude of the shear stress



and the properties of the riverbed and armor layer deposited onto the riverbed. The shear stress is mainly a function of fluid velocity and water level at the streambed (Hubbs et al., 2007; Stuyfzand et al., 2006). Shear stress values have been reported to range between 1 to 100 N/m$^2$ as typical for river streambeds, considering a value of 20 N/m$^2$ as reasonable for the design of a RBF (Hubbs, 2006b). Schubert (2002) stated an approximate average shear stress of 10 N/m$^2$ in the Lower Rhine River

region at the Flehe waterworks. Hubbs (2006b) reported a minimum shear stress (during low flow conditions) of 0.2 N/m$^2$ and maximum shear stress of 9.16 N/m$^2$ (during high flow conditions) in the Ohio River at Louisville, Kentucky. While flood events may stimulate riverbed renewal by streambed scouring as the result of shear forces, low flow periods may promote sedimentation of suspended solids at the riverbed (Levy et al., 2011; Stuyfzand et al., 2006). However, Schubert (2002) stated that flood events might also induce riverbed clogging due to the higher concentration of suspended solids and a higher gradient

between the river level and the water table of the aquifer.

Scouring or self-cleansing capacity of RBF systems is commonly assessed in terms of critical shear stress that depends on riverbed particle characteristics (considering its critical shear stress) and the shear stress exerted by the river water velocity. Viscosity and density of the fluid contribute to shear stress forces (Hubbs et al., 2007), but these properties are expected to be constant in time in tropical rivers (Lewis Jr., 2008). The velocity of the fluid at the streambed is a function of stream surface

slope and water level, and resistance to flow transmitted by the streambed. These parameters vary in time and place, determining the sediment transport capacity on the surface of the streambed (Hubbs et al., 2007).

Erosion and deposition behave dissimilarly for cohesive and non-cohesive sediments (Winterwerp and van Kesteren, 2004). Ahmad et al. (2011) experimentally studied the critical shear stress using sand and different mud mixtures, stating an increase in the critical shear stress by a factor of 1.5 for a mixture with a mud fraction of 50% in comparison to only sand. For non-

cohesive sediments, when bed shear stress is greater than the critical shear stress, erosion and deposition occur simultaneously (Krishnappan, 2007). By contrast, for cohesive sediments, erosion and deposition do not act simultaneously for all shear stress conditions due to electrochemical and biological processes binding the cohesive particles to the riverbed. Armor layers made up from deposition of cohesive materials carried by the rivers will increase their resistance to erosive processes, resulting in higher shear stresses to move the sediments deposited on the riverbed. In addition, the shear stress for deposition of cohesive

sediments is different from the shear stress for erosion (Krishnappan, 2007). As stated by Berlamont et al. (1993), the critical shear stress for deposition is usually in the range of 0.05-0.2 N/m$^2$, whilst for erosion is in the range of 0.1-2 N/m$^2$. Moreover, cohesive sediments consolidate over time when deposited on a bed, altering the critical shear stress for erosion through compaction (Krishnappan and Engel, 1994), while their bulk densities tend to increase as a function of depth and time (Lick, 2008). Jepsen et al. (1997) studied the changes in bulk density as a result of depth and consolidation time in Detroit and Fox

Rivers, and Santa Barbara slough, finding that, although different bulk densities were obtained among the locations, the density variation trends were similar. Thus, an increase of the bulk density by depths up to 0.2%/cm in the river sediments, and 0.7%/cm in the slough sediments. Regarding to consolidation time, increases up to 0.1%/d in the river sediments, and up to 0.3%/d in the slough sediments. Therefore, bed age or consolidation time might play an important role in critical shear stress



values and erosion rates for deposited cohesive sediments (Droppo and Amos, 2001; Jepsen et al., 1997; Krishnappan and Engel, 1994; Stone et al., 2008; Valentine et al., 2014).

## 4 Discussion about the applicability of RBF in Colombia

It may be concluded that RBF is a technology appropriate for use in highly turbid and contaminated surface rivers in Colombia
(Gutiérrez et al., 2016), due to its capacity to remove a high variety of pollutants linked to the influence of the high suspended sediment loads carried by the rivers. As a consequence of the suspended sediments, cake formation on the riverbed and clogging of the aquifer may occur (Caldwell, 2006), contributing to the removal of most dissolved and suspended contaminants (Ray, 2002a). In addition, a good water quality can be obtained at the abstraction wells, requiring only a few additional treatment steps for the production of drinking water (Singh et al., 2010; Sprenger et al., 2014; Thakur and Ojha, 2010).
In Colombia, nowadays, conventional surface water treatment plants (coagulation-flocculation-sedimentation-filtration-chlorination) are used for supplying drinking water. RBF as an alternative pre-treatment step may provide an important reduction of chemicals' consumption, considerably simplifying the operation of the existing treatment processes. It is expected that employing RBF in communities where the conditions are appropriate for its implementation (e.g. located in an alluvial formation and close to a river,) will lead to considerable improvements in source water quality. Mainly, improvements due to
the removal of turbidity, and pathogens, and to a lesser extent inorganics, organic matter and micro-pollutants are expected. Furthermore, in Colombia, shock loads of pollutants commonly lead to shutdowns of water treatment plants until the peak has passed (Gutiérrez et al., 2016; Pérez-Vidal et al., 2012). RBF has the potential to mitigate shock loads (Schmidt et al., 2003) thus leading to the prevention of shutdowns of water treatment plants.

RBF thus typically results in fewer environmental impacts than conventional surface water treatment. The environmental
benefits can mainly be attributed to its considerable reductions in chemical usage and sludge production. Likewise, the eliminations of surface water intake structures may have a positive effect on the surrounding aquatic environment. However, the high sediment loads contained in many Colombian rivers may lead to some negative environmental impacts with the use of RBF, mainly associated to changes in vital aquatic habitats caused by riverbed clogging (Kendy and Bredehoeft, 2007).

The suspended sediments, responsible for the clogging processes, may on the one hand be favorable in the improvement of the
water quality mainly due to the strengthening of cake filtration and deep bed filtration processes. On the other hand, the formed cake layer must be balanced by scouring in order for an RBF system to be sustainable. Therefore, clogging and self-cleansing issues must be studied in greater depth to assess the use of RBF technology in highly turbid waters, because they may affect the abstraction capacity yield as well as the development of different redox zones for efficient contaminant removal.

Finally, in the design of a RBF system, a balance between the water quality and the production capacity must be sought.
Greater removal efficiencies may be achieved with increased travel distances (residence time), yet there is an inevitable trade-off between the ability to supply large flows and the decreased water quality in the abstraction wells. For a RBF system to be



sustainable, the infiltration rate must remain high enough throughout the river-aquifer interface in order to provide the water quantity needed, and the residence time of the contaminants must be enough to ensure an adequate water quality.

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

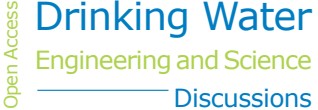



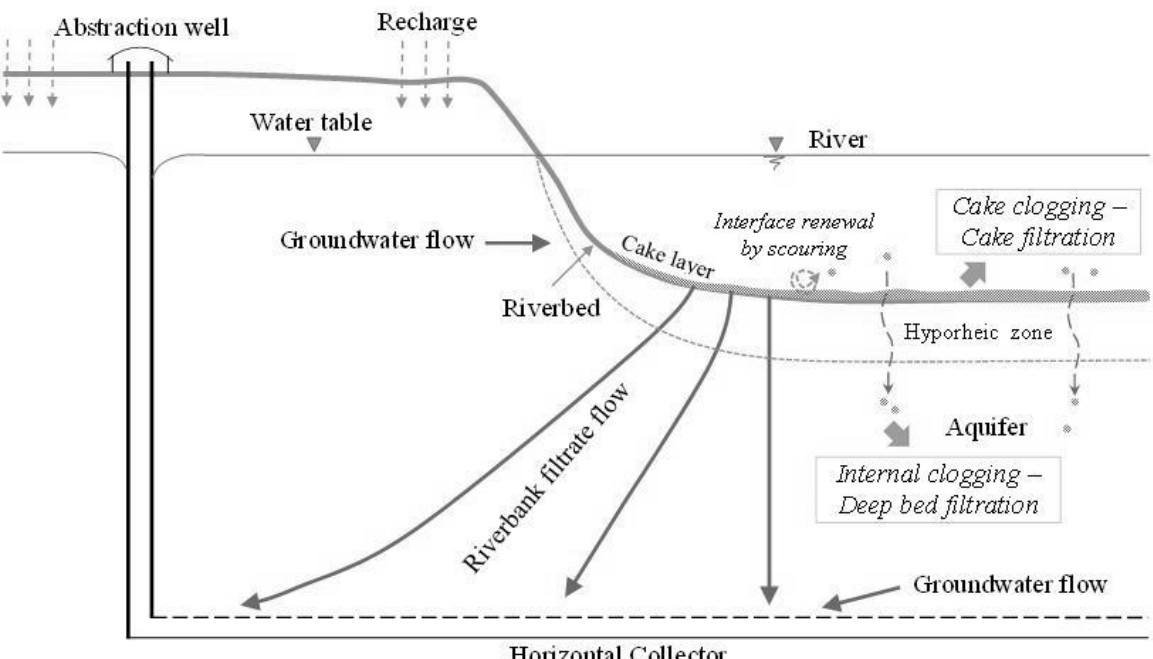

**Figure 1: General representation of a horizontal RBF system**

**Table 1: Turbidity removal at bank filtration sites with highly turbid raw water sources**

| Bank filtration site | Distance from source water (m) V: vertical H: horizontal | Source water (maximum turbidity, NTU) | Bank filtration system (maximum turbidity, NTU) | Turbidity removal (%) |
|---|---|---|---|---|
| Pant Dweep Island at Haridwar, India (Thakur and Ojha, 2010) | 180 (V) | 2,500 | – | ±99.9 |
| Uttarakhand Jal Sansthan at Haridwar, India (Dash et al., 2010) | 320 (V) | 200 | 0.6 | 99.70 |
| Indiana-American Water at Jeffersonville, USA (Weiss et al., 2005) | 177 (V) 30 (V) | 661 | 1.1 1.5 | 99.83 99.77 |



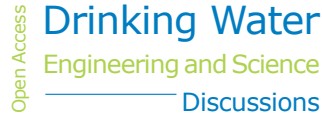

| Bank filtration site | Distance from source water (m) V: vertical H: horizontal | Source water (maximum turbidity, NTU) | Bank filtration system (maximum turbidity, NTU) | Turbidity removal (%) |
|---|---|---|---|---|
| Indiana-American Water at Terre Haute, USA (Weiss et al., 2005) | 24 (H) 122 (V) | 1,761 | 0.27 0.41 | 99.98 99.98 |
| Missouri-American Water at Parkville, USA (Weiss et al., 2005) | 37 (V) 37 (V) | 1,521 | 3.8 2.7 | 99.75 99.82 |
| Louisville at Kentucky, USA (Wang, 2003) | 23 (V) 24 (H) | 599 | ±0.8 0.69 | ±99.8 99.88 |