# Peer review of "Riverbank filtration for treatment of highly turbid Colombian rivers"

_Drinking Water Engineering and Science, 2017_

## Referee Comment (RC1) · Anonymous Referee #1 · 1 Mar 2017

Reviewer's comments on manuscript no: dwes-2017-10 Title: Riverbank filtration for treatment of highly turbid Colombian rivers

General comments This review discusses 'riverbank filtration for treatment of highly turbid Columbian waters'. This is a very important contribution which makes use of banks of rivers to purify water. Utilization of river bank filtration still remains a thorny issue since the removal efficiency depends not only o the contaminants, but also the hydraulic and chemical characteristics, local discharge conditions and biochemical processes. These is need thus need for increased knowledge and understanding of the practicability of riverbank filtration for highly polluted waters. I have specific comments that will help improve the manuscript.

Specific comments Abstract ïĂ∎ Lines 6-8-the first sentence is not clear. It is rather too

long and compounded. ïĂ■ I suggest that the entire abstract should be re-written to give the reader a clear picture of the problem that led the authors to come up with the review, why is the problem a problem, what were the objectives of the review, summary of methods used, summary of major findings of the review and conclusions as well as recommendations in logical sequence.

Introduction ïĂ■ Page 2, line 16-authors ought to give examples of the 'mutagenic compounds' and 'certain organic and inorganic micropollutants' being referred to here. This is a review article, there is no need to speculate. ïĂ■ What are the reported removal efficiencies of the 'mutagenic compounds' and 'certain organic and inorganic micropollutants' by riverbank filtration? ïĂ■ Page 2, line 17-which are some of these 'specific micropollutants that remain mobile, and why? ïĂ■ Page 2, lines 20-21, what is the feasibility of using riverbank filtration as a pretreatment method considering rate of productivity and revel time of water along flow paths? ïĂ■ Page 2, lines 24-25-how high were the reported turbidity values? There is need for actual figures here. What do authors mean by 'contamination events'? ïĂ■ Page 6-lines 23-25- out of 247 micro-pollutants only 14 were completely removed, what were the removal efficiencies of the remaining ones? ïĂ■ Page 6-lines 23-25-what were the chances of the 14 compounds reported to have been completely removed undergoing transformations and forming degradates? 3.6 years looks like a longer period considering increased demand for water of good quality and quantity? ïĂ■ How about adsorption and complexation of the pollutants to the soil along flow paths which could result into soil pollution and groundwaters at the expense of purifying surface water using riverbank filtration. Consider investigating chances of creating another problem at the expense of solving other problems

Discussion ïĂ■ Table 1-What were the chances of using one sample of water to compare the maximum turbidity levels in Source water and river bank filtration system? ïĂ■ Was there any relationship between orientation, slope, type of soil within the riverbanks, travel time and turbidity removal percentages? What other factors need to be carefully considered in order to improve efficiency and productivity of the riverbank

filtration systems. ïĂ■ Page 10, line 29-very good point, but are there any suggestions to that effect? ïĂ■ What is the relationship between the removal period/residence time of contaminants by river bank filtration and sustainability considering the ever increasing demand for water worldwide? Is riverbank filtration a feasible option to solve water quality changes at a larger scale? ïĂ■ How does use of riverbank filtration (whose efficiency is site and substance specific) compare with other equally important methods of water purification such as use of sand filtration, activated carbon etc that are used in the treatment of highly turbid and polluted waters? There is need for a discussion and comparative assessment on the productivity/production capacity and performance of the riverbank filtration method versus other methods in removal of turbidity, organic and in organic compounds, otherwise making conclusions out of this consideration is somehow questionable.

Please also note the supplement to this comment:
http://www.drink-water-eng-sci-discuss.net/dwes-2017-10/dwes-2017-10-RC1-supplement.pdf

---

## Referee Comment (RC2) · Anonymous Referee #2 · 20 Mar 2017

Riverbank filtration for treatment of highly turbid Colombian rivers Manuscript number: DWES-2017-10

Work on riverbank filtration (RBF) for treatment of highly turbid water is interesting as reports have shown that RBF can be used as an alternative pre-treatment step in order to reduce the use of chemicals. The authors report on considerable poor quality of many Colombian surface water and suggest RBF as an alternative treatment process. This can be good for large scale treatment not only for Colombian surface waters but also highly turbid water from other countries. The review is well written and informative. Having said that, I have the following comments for the authors. Page 4, line 4: Please explain the role of the schmutzdecke layer in biological treatment? Section 4: Please highlight the potential challenges in the application of RBF in conventional surface water treatment plants in Colombia. With regards to construction, maintenance

and operational costs, would the use of RBF as pretreatment be cost-effective over a long term? What is the competitiveness of RBF to slow sand filtration (SSF)? Please highlight the limitations of the current systems used in treatment of surface water and suggest which treatment steps RBF can replace (or eliminate) if incorporated in the current water purification plants. What would be the willingness (acceptance) of surface water treatment plants to incorporate RBF in their current treatment chain taking into account cost-effectiveness and amount of space required compared to membrane bioreactor (MBR)?

---

## Referee Comment (RC3) · Anonymous Referee #3 · 23 Mar 2017

Manuscript title: Riverbank filtration for treatment of highly turbid Colombian rivers General comment This work discusses various studies where performed using river bank filtration (RBF) technology was used to purify surface water and has the inherent advantage of reducing the utilization of chemicals during water treatment. The presented studies showed effective removal of dissolved and suspended sediments and removal pollutants and pathogens. It also, to a limited extent explains the various purification mechanisms involved during RBF. The manuscript is well written and chronologically presented and presents areas for furthers studies that could improve the fundamental understanding of river bank filtration. However, I still think the discussion could have been made more informative by including details in some of the discussed processes and removal mechanisms as indicated in the specific comments.

Specific comments 1. The abstract needs to be revised to give brief highlights of

all the findings discussed in this review not just limited to suspended solid removal. 2. Page 4 lines 11-15 reports that heavy metals are mainly removed through ion exchange, is it that extensive as reported and what exactly could be possible for this property (which part of the filter bed does this occurs). I understand the bed is mainly made of sand which mainly utilizes size exclusion as a removal mechanism. 3. How does variation in seasons influence the RBF performance and other influential factors such as clogging (due to high load of suspended solids). Do the authors have some information/comments on this? 4. Page 7, section 3.1 the reported clogging was is it only a function of the deposition of suspended colloidal matter? What about dissolved organic micro-molecules such as polysaccharides or extracellular polymeric substances? 5. Section 2.6 summarizes some interesting findings on micro pollutant removal, the discussion could have been clearer if the mechanisms of removal were also discussed because I don't believe size exclusion played a significant role in removal. Generally micro-pollutants are removed through three possible routes: charge interactions (electrostatic interactions), pollutant-substrate interactions (hydrophobic/hydrophilic interactions) and non-electrostatic interactions (acid-base interactions). Can the authors comment on this? 6. The impressive removal of some micro-pollutants; could it have been due to the microbial degradation/microbial activity? 7. And if it is degradation, what intermediates/metabolites are formed? Did these studies determine the degradation products.

Please also note the supplement to this comment:
http://www.drink-water-eng-sci-discuss.net/dwes-2017-10/dwes-2017-10-RC3-supplement.pdf
* * *

---

## Author Comment (AC1)

**Response to Anonymous Referee #1**

First of all, thank you for your comments to the discussion of our manuscript.
The suggestions made by the reviewer were considered, and the respective changes were applied to the manuscript as follows:

Abstract

The abstract was modified as suggested by the reviewer.

The poor water quality of many Colombian surface waters, forces for seeking alternative, sustainable treatment solutions with the ability to manage peak pollution events and to guarantee an uninterrupted provision of safe drinking water to the population. This review assesses the potential of using riverbank filtration (RBF) for the highly turbid and contaminated waters in Colombia emphasizing on water quality improvement and the influence of clogging by suspended solids. The suspended sediments may on the one hand be favorable in the improvement of the water quality mainly due to the strengthening of cake filtration and deep bed filtration processes. On the other hand, the formed cake layer must be balanced by scouring in order for an RBF system to be sustainable without loss of hydraulic capacity. In general, RBF seems to be a technology appropriate for use in highly turbid and contaminated surface rivers in Colombia, where improvements due to the removal of turbidity, and pathogens, and to a lesser extent inorganics, organic matter and micro-pollutants are expected. RBF has the potential to mitigate shock loads thus leading to the prevention of shutdowns of surface water treatment plants. In addition, RBF, as an alternative pre-treatment step, may provide an important reduction of chemicals' consumption, considerably simplifying the operation of the existing treatment processes. RBF may be considered as a feasible option to solve water quality changes at a larger scale. However, clogging and self-cleansing issues must be studied deeper in the context of these highly turbid waters, evaluating the potential loss of abstraction capacity yield as well as the development of different redox zones for efficient contaminant removal.

Introduction

- P2, 16 authors ought to give examples of the 'mutagenic compounds' and 'certain organic and inorganic micropollutants' being referred to here. This is a review article, there is no need to speculate. What are the reported removal efficiencies of the 'mutagenic compounds' and 'certain organic and inorganic micropollutants' by riverbank filtration?
R/ "In addition, RBF has demonstrated an ability to decrease mutagenic compounds, including naproxen, gemfibrozil and ibuprofen (Hoppe-Jones et al., 2010; Schubert, 2003) and to remove organic and inorganic micro-pollutants, such as sulfamethoxazole and propranolol (Bertelkamp et al., 2014; Hamann et al., 2016; Schmidt et al., 2003)."

- P2, 17 which are some of these 'specific micropollutants that remain mobile, and why?

R/ *"However, it has also been found that specific micro-pollutants such as carbamazepine and EDTA remain mobile, showing a persistent behavior even after 3.6 years of travel time (Hamann et al., 2016). The persistence is mainly driven by the very low reactive and sorptive characteristics of these compounds (Scheytt et al., 2006)."*

- P2, 20-21 what is the feasibility of using riverbank filtration as a pretreatment method considering rate of productivity and travel time of water along flow paths?
R/ "Although RBF has shown to be highly effective in the removal of many contaminants, it must mainly be considered as a pre-treatment method, which needs to be combined with a certain post-treatment (Cady et al., 2013; Dash et al., 2008; Kuehn and Mueller, 2000; Singh et al., 2010). A balance between the water quality and the production capacity must be considered, where greater removal efficiencies are achieved by increasing travel distances, but decreasing the rate of productivity."

- P2, 24-25 how high were the reported turbidity values? There is need for actual figures here. What do authors mean by 'contamination events'?
R/ "However, in the last decades, turbidity and contamination events in surface waters have become a serious concern in Colombia for guaranteeing safe drinking water (Gutiérrez et al., 2016; Universidad del Valle and UNESCO-IHE, 2008). Fast urbanization, the lack of integration between water management and spatial planning, and inappropriate land use are identified as the main causes for the progressive deterioration of the surface water (IDEAM, 2015; van der Kerk, 2011; Universidad del Valle and UNESCO-IHE, 2008). Figure 2 illustrates the monthly turbidity variation in percentiles in the Cauca River (Cali, Colombia) for years 2008-2013 (EMCALI, personal communication, August 21, 2015). High turbidity events in the Cauca River lead to the intake shutdowns in the main water treatment plant (Puerto Mallarino WTP) of the city of Cali, reporting turbidity peaks up to 10,000 NTU (Figure 2). The decrease in the DO concentrations in the Cauca River is used as an indicator of high pollution peaks. It typically drops after heavy rainfalls with the increase of organic matter concentrations (CVC and Universidad del Valle, 2004)."

[Figure]

Figure 2. Turbidity percentile values in Cauca River, Colombia, during years 2008-2013

- P6, 23-25 out of 247 micro-pollutants only 14 were completely removed, what were the removal efficiencies of the remaining ones?. What were the chances of the 14 compounds reported to have been completely removed undergoing transformations and forming degradates? 3.6 years looks like a longer period considering increased demand for water of good quality and quantity? How about adsorption and complexation of the pollutants to the soil along flow paths which could result into soil pollution and groundwaters at the expense of purifying surface water using riverbank filtration. Consider investigating chances of creating another problem at the expense of solving other problems

R/ As stated by Hamann et al. (2016), 247 micro-pollutants were analyzed during 14 years, but only 29 were selected for the detailed fate analysis due to different reasons explained by the authors in the manuscript. Considering that, the sentence was reworded to make it clearer to the reader. All the comments made by the reviewer were considered and included, as follows: "Hamann et al. (2016) analyzed the fate of 29 micro-pollutant compounds in a RBF system considering a travel time up to 3.6 years, finding complete removal of 14 compounds (2-naphthalene sulfonate, 2,6-NDS, amidotrizoic acid, AMPA, aniline, bezafibrate, diclofenac, ibuprofen, iohexol, iomeprol, iopromide, ioxitalamic acid, metoprolol and sulfamethoxazol) due to retardation and degradation processes as supported from numerical modeling. In addition, some compounds were partially removed (triglyme, iopamidol, diglyme, 1,3,5-naphthalene trisulfonate, 1,3,6-naphthalene trisulfonate), with removal efficiencies ranging from approximately 60 to 90%, based on the highest concentrations measured in both the Lek River and observation well (906 m from river, 3.65 years travel time). Only 10 compounds were fully persistent during the subsurface passage in the RBF system (1,4-dioxan, 1,5-naphthalene

disulfonate (1,5-NDS), 2-amino-1,5-NDS, 3-amino-1,5-NDS, AOX, carbamazepine, EDTA, MTBE, toluene and triphenylphosphine oxide). The authors do not differentiate between biodegradation and sorption; where adsorption, ion-pair formation and complexation of pollutants to the soil may lead to soil pollution (Bradl, 2004)."

Discussion

- Table 1-What were the chances of using one sample of water to compare the maximum turbidity levels in Source water and river bank filtration system? Was there any relationship between orientation, slope, type of soil within the riverbanks, travel time and turbidity removal percentages? What other factors need to be carefully considered in order to improve efficiency and productivity of the riverbank filtration systems.
R/ The Table 1 was built based on secondary information. The turbidity removal percentages presented were computed from the maximum turbidity values reported in both surface and bank filtered water, considering that the authors intended to show the behavior of RBF under critical conditions such as suspended sediments content (expressed as turbidity). In order to improve the Table 1, travel time and aquifer material were included. Other factors were considered such as slope and streambed material; however, that information was not reported in the cited articles.

Hence, the reviewer comments were considered in the Table 1 and the paragraph as follows:

"The RBF system configuration (i.e. vertical, horizontal) does not govern the suspended solids removal efficiency as observed in Table 1, since it is not a function of the travel/contact time. The texture of the streambed, however, influences the media clogging (Hubbs et al., 2007; Stuyfzand et al., 2006), where external clogging (cake layer formation) enhances the removal capacity of fine sediments contained in the water (Veličković, 2005). The removal efficiency of suspended solids is concentration dependent (Fallah et al., 2012; Thakur and Ojha, 2010); the higher the suspended solids concentration, the faster the cake formation and therefore the higher the turbidity removal capacity. Although, no studies have quantified the role of concentration on entrapment, the critical particle concentration where the porous media gets clogged has been determined to be dependent on the ratio of void size to particle size (Sen and Khilar, 2006). As reported by Sen and Khilar (2006), the critical particle concentration increased from 0.35% to 9% when the ratio of bead size to particle size was increased from 12 to 40. Therefore, the removal efficiency of suspended solids is a function of both the filtering media characteristics (streambed and particle sizes of the aquifer), and the water quality in terms of suspended particle size and concentration"

**Table 1: Turbidity removal at bank filtration sites with highly turbid raw water sources**

| Bank filtration site | Pant Dweep Island at Haridwar, India (Dash et al., 2010) (Thakur and Ojha, 2010) | Indiana-American Water at Jeffersonville, USA (Weiss et al., 2005) | Indiana-American Water at Terre Haute, USA (Weiss et al., 2005) | Missouri-American Water at Parkville, USA (Weiss et al., 2005) | Louisville at Kentucky, USA (Wang, 2003) |
|---|---|---|---|---|---|

| Distance from source water
V: vertical
H: horizontal | 320 m (V)
108 m (V) | 177 m (V)
30 m (V) | 24 m (H)
122 m (V) | 37 m (V)
37 m (V) | 23 (V)
24 (H) |
|---|---|---|---|---|---|
| Travel time (d) | 420-510
32.5 | 13-19
3-5 | NA | NA | 2-5 |
| Source water (maximum turbidity, NTU) | 200
2,500 | 661 | 1,761 | 1,521 | 599 |
| Bank filtration system (maximum turbidity, NTU) | 0.6
-- | 1.1
1.5 | 0.27
0.41 | 3.8
2.7 | ±0.8
0.69 |
| Turbidity removal (%) | 99.7
±99.9 | 99.83
99.77 | 99.98
99.98 | 99.75
99.82 | ±99.8
99.88 |
| Aquifer material | Sand, clayey and silty sands | Clay, fine and medium sands, coarse gravels | Medium and fine sands underlain by coarser sand and gravel | Fine to coarse sand, gravel, and boulder deposits with intermixed layers of clay and silt overlying consolidated shale and limestone | Sand and gravel with silt and clay |

NA – Not available

- P10, 29 very good point, but are there any suggestions to that effect? What is the relationship between the removal period/residence time of contaminants by river bank filtration and sustainability considering the ever increasing demand for water worldwide? Is riverbank filtration a feasible option to solve water quality changes at a larger scale?
R/ To give response to the reviewer´s comments, the paragraph was reworded as follows:
"Finally, in the design of a RBF system, a balance between the water quality and the production capacity must be sought. Greater removal efficiencies may be achieved with increased travel distances (residence time), yet there is an inevitable trade-off between the ability to supply large flows and the decreased water quality in the abstraction wells. The longer the travel distance the higher fraction of groundwater extracted from storage in the aquifer; and therefore, the lower the extraction capacity of the system (de Vet et al., 2010). For a RBF system to be sustainable, the infiltration rate must remain high enough throughout the river-aquifer interface in order to provide the water quantity needed, and the residence time of the contaminants must be enough to ensure an adequate water quality. Nonetheless, even with shorter residence times, the abstracted water will have better characteristics than the raw water, making further treatment steps such as coagulation, flocculation and sedimentation redundant. Therefore, RBF may be considered as a feasible option to solve water quality changes at a larger scale."

- How does use of riverbank filtration (whose efficiency is site and substance specific) compare with other equally important methods of water purification such as use of sand filtration, activated carbon etc that are used in the treatment of highly turbid and polluted waters? There is need for a discussion and comparative assessment on the productivity/production capacity and performance of the riverbank filtration method versus other methods in removal of turbidity,

organic and inorganic compounds, otherwise making conclusions out of this consideration is somehow questionable.

R/ A paragraph describing the problem in using conventional WTPs in Colombia and a comparative assessment of combinations of trains has been included as follows:

As stated by Gutiérrez et al. (2016), in Colombian WTPs the operation and maintenance and sludge disposal are the main processes leading to costly water production. The costs are linked to chemical usage, sludge production and its treatment. A brief comparison of robust drinking water technologies in removal of turbidity, pathogens and the chemical contaminants discussed during this review is realized based on the analysis conducted by Hubbs et al. (2003) and Ray and Jain (2011). Slow sand filtration, with pre-treatment, is mainly suitable for small to medium sized communities, whereas RBF and conventional WTP can be suitable for small to very large communities (Ray and Jain, 2011). RBF is suitable for highly contaminated rivers, able to match conventional treatments including advanced technologies such as ozone, ultraviolet light or granular activated carbon for pesticides' removal. Although using a conventional train such as coagulation – sedimentation – filtration – activated carbon filtration – disinfection $(O_3/UV/H_2O_2/Cl_2)$ and an alternative train such as RBF – aeration – filtration – activated carbon filtration – disinfection $(O_3/UV/H_2O_2/Cl_2)$ may produce similar water qualities, there are differences in the production costs. The use of RBF leads to savings of chemical dosing, sludge handling and filter backwashing. As reported by Sharma and Amy (2009), the conversion from a conventional WTP to a process including a RBF system may reduce the operational costs up to 50%. Moreover, the sedimentation step may be skipped, and advanced removal of pathogens is no longer needed. As reported by Dusseldorp (2013), after anaerobic river bank filtrate is extracted in a WTP train in the Netherlands, water is pre-treated with reverse osmosis prior to conventional treatment steps of sand filtration, granular activated carbon and UV disinfection, in order to use in combination with membrane filtration avoiding ultrafiltration and biofouling. RBF has the advantage over the other assessed technologies of dampening shock loads and peaks, which is a need in rivers with extreme variable water qualities such as the Colombian rivers (e.g. Cauca River, Figure 2).

---

## Author Comment (AC2)

**Response to Anonymous Referee #2**

Many thanks to the reviewer for the comments to improve the manuscript.
All the suggestions have been addressed, and the changes were applied in the manuscript as follows:

- P4, 4 Please explain the role of the schmutzdecke layer in biological treatment?
R/ "In the first centimeters of the riverbed a fine sediments' layer is formed, also known as cake layer. The cake layer is called schmutzdecke if a highly active biological layer is involved (Hiscock and Grischek, 2002; Unger and Collins, 2006). A certain degree of clogging in the riverbed is preferred since it can be favorable for water quality improvement (Ray and Prommer, 2006), due to the augmentation of traveling times, particulate removal and the richness of processes occurring in the schmutzdecke (Hiscock and Grischek, 2002; Schmidt et al., 2003; Unger and Collins, 2006). Jüttner (1995) determined e.g. that the schmutzdecke and upper layers were responsible for most of the elimination of volatile organic carbon, and Dizer et al. (2004) concluded that this layer is extremely efficient in eliminating viruses. Maeng et al. (2008) found that the 50% of the total dissolved organic matter removal in a simulated RBF system occurred in the first few centimeters of infiltration surface due to the biological activity in the developed biomass. In the schmutzdecke layer, the removal of organic matter, pathogens and chemicals occurs by predation, scavenging and metabolic breakdown mechanisms (Haig et al., 2011). A cake layer, mainly composed of organic and/or clay constituents, may also enhance the sorption of pollutants onto its surface (Li et al., 2003)."

- Section 4:
Please highlight the potential challenges in the application of RBF in conventional surface water treatment plants in Colombia. With regards to construction, maintenance and operational costs, would the use of RBF as pretreatment be cost-effective over a long term? What is the competitiveness of RBF to slow sand filtration (SSF)? Please highlight the limitations of the current systems used in treatment of surface water and suggest which treatment steps RBF can replace (or eliminate) if incorporated in the current water purification plants. What would be the willingness (acceptance) of surface water treatment plants to incorporate RBF in their current treatment chain taking into account cost-effectiveness and amount of space required compared to membrane bioreactor (MBR)?
R/ Two subheading were added. One subheading was focused in the comparison of water treatment technologies, whilst the other subheading was addressed to discuss the challenges of applying RBF in conventional WTPs. The information included reads as follows:

4.1 Comparative assessment of water treatment technologies
In Colombia, nowadays, conventional surface water treatment plants (coagulation-flocculation-sedimentation-filtration-chlorination) are used for supplying drinking water. As stated by Gutiérrez et al. (2016), in Colombian WTPs the operation and maintenance and sludge disposal are the main processes leading to costly water production. The costs are linked to chemical usage, sludge production and its treatment. A brief comparison of robust drinking water

technologies in removal of turbidity, pathogens and the chemical contaminants discussed during this review is realized based on the analysis conducted by Hubbs et al. (2003) and Ray and Jain (2011). Slow sand filtration, with pre-treatment, is mainly suitable for small to medium sized communities, whereas RBF and conventional WTP can be suitable for small to very large communities (Ray and Jain, 2011). RBF is suitable for highly contaminated rivers, able to match conventional treatments including advanced technologies such as ozone, ultraviolet light or granular activated carbon for pesticides' removal. Although using a conventional train such as coagulation – sedimentation – filtration – activated carbon filtration – disinfection ($O_3$/UV/$H_2O_2$/$Cl_2$) and an alternative train such as RBF – aeration – filtration – activated carbon filtration – disinfection ($O_3$/UV/$H_2O_2$/$Cl_2$) may produce similar water qualities, there are differences in the production costs. The use of RBF leads to savings of chemical dosing, sludge handling and filter backwashing. As reported by Sharma and Amy (2009), the conversion from a conventional WTP to a process including a RBF system may reduce the operational costs up to 50%. Moreover, the sedimentation step may be skipped, and advanced removal of pathogens is no longer needed. As reported by Dusseldorp (2013), after anaerobic river bank filtrate is extracted in a WTP train in the Netherlands, water is pre-treated with reverse osmosis prior to conventional treatment steps of sand filtration, granular activated carbon and UV disinfection, in order to use in combination with membrane filtration avoiding ultrafiltration and biofouling. RBF has the advantage over the other assessed technologies of dampening shock loads and peaks, which is a need in rivers with extreme variable water qualities such as the Colombian rivers (e.g. Cauca River, Figure 2).

4.2 Potential challenges in the application of RBF in conventional surface water treatment plants in Colombia

RBF as an alternative pre-treatment step may provide an important reduction of chemicals' consumption, considerably simplifying the operation of the existing treatment processes. It is expected that employing RBF in communities where the conditions are appropriate for its implementation (e.g. located in an alluvial formation and close to a river,) will lead to considerable improvements in source water quality. Mainly, improvements due to the removal of turbidity, and pathogens, and to a lesser extent inorganics, organic matter and micro-pollutants are expected. Furthermore, in Colombia, shock loads of pollutants commonly lead to shutdowns of water treatment plants until the peak has passed (Gutiérrez et al., 2016; Pérez-Vidal et al., 2012). RBF has the potential to mitigate shock loads (Schmidt et al., 2003) thus leading to the prevention of shutdowns of water treatment plants.

During the application of RBF in conventional surface WTPs in Colombia, many of the treatment processes currently employed could be varied or even removed completely, leading to simpler plant operation and control. In the specific case of the Puerto Mallarino WTP in Cali, Colombia, RBF would replace all current pre-treatment process steps occurring in the grit chamber, rapid mix chamber, and the flocculation and settling clarifiers (Gutiérrez et al., 2016). Chemical doses could be reduced in all remaining processes, but an additional requirement for aeration directly after well extraction may be needed. However, this would only be necessary in the instance that the RBF filtrate had become anaerobic during soil passage. Because of the process changes a stable inflow quality (turbidity, temperature, pH and electrical conductivity) means that the plant will operate under more stable conditions, thereby increasing plant

efficiency and effluent quality. RBF well operation and control is much simpler than the existing treatment steps, which currently require continual adjustment to ensure smooth plant operation according to any changes in raw water quality. Additionally, a complete reduction in the sludge produced by the grit chambers and clarifiers would be achieved.

---

## Author Comment (AC3)

**Response to Anonymous Referee #3**

Thank you for your comments to improve our manuscript.
The suggestions made by the reviewer were considered, and the respective changes were applied to the manuscript as follows:

1. The abstract needs to be revised to give brief highlights of all the findings discussed in this review not just limited to suspended solid removal.

The abstract was modified as suggested by the reviewer.

The poor water quality of many Colombian surface waters, forces for seeking alternative, sustainable treatment solutions with the ability to manage peak pollution events and to guarantee an uninterrupted provision of safe drinking water to the population. This review assesses the potential of using riverbank filtration (RBF) for the highly turbid and contaminated waters in Colombia emphasizing on water quality improvement and the influence of clogging by suspended solids. The suspended sediments may be favorable in the improvement of the water quality, but may reduce the production yield capacity. The cake layer must be balanced by scouring in order for an RBF system to be sustainable. The infiltration rate must remain high enough throughout the river-aquifer interface in order to provide the water quantity needed, and the residence time of the contaminants must be enough to ensure an adequate water quality. In general, RBF seems to be a technology appropriate for use in highly turbid and contaminated surface rivers in Colombia, where improvements due to the removal of turbidity, and pathogens, and to a lesser extent inorganics, organic matter and micro-pollutants are expected. RBF has the potential to mitigate shock loads thus leading to the prevention of shutdowns of surface water treatment plants. In addition, RBF, as an alternative pre-treatment step, may provide an important reduction of chemicals' consumption, considerably simplifying the operation of the existing treatment processes. However, clogging and self-cleansing issues must be studied deeper in the context of these highly turbid waters, evaluating the potential loss of abstraction capacity yield as well as the development of different redox zones for efficient contaminant removal.

2. P4, 11-15 the sentence reports that heavy metals are mainly removed through ion exchange, is it that extensive as reported and what exactly could be possible for this property (which part of the filter bed does this occurs). I understand the bed is mainly made of sand which mainly utilizes size exclusion as a removal mechanism.

R/ Additional information was included in order to make it clearer to the reader. Therefore, the modified paragraph reads as follows:

"The removal of heavy metals from source water during subsurface passage mainly occurs by sorption, precipitation and ion exchange processes, which depend on the content of inorganic and organic compounds in the aquifer and contact time (Bourg et al., 1989; Hülshoff et al., 2009). Under aerobic conditions, heavy metals removal is mainly attributed to ion exchange processes

at negatively loaded surfaces (Schmidt et al., 2003). The presence of negatively charged surfaces (e.g. clayey and/or organic sediments) and amorphous ferric and alumina oxides provide exchange sites for binding trace heavy metals (Foster and Charlesworth, 1996; Salomons and Förstner, 1984). As contact time is a critical parameter affecting the fate of most heavy metals, the removal of such compounds by ion exchange processes mainly occurs in the hyporheic zone and the flow path between the river and the abstraction well (Hülshoff et al., 2009; Stuyfzand, 2011)."

3. How does variation in seasons influence the RBF performance and other influential factors such as clogging (due to high load of suspended solids). Do the authors have some information/comments on this?

R/ The paragraph in P7, lines 21-24 was completed in order to give a proper response to the reviewer´s question. Thus:

…Clogging has been identified as the major contributor to the long-term decay of RBF yield (Hubbs et al., 2007), but there is a lack of understanding of the exact factors that affect clogging (Caldwell, 2006; Hubbs et al., 2007; Schubert, 2006a; Stuyfzand et al., 2006). Hubbs et al. (2007) reported a decrease in the specific capacity of the wells up to 67% of its initial level in the first 4-year period of operation. Most of the reduction took place within the first year due to riverbed clogging in the vicinity of the well. Clogging is time dependent and is a function of bed material (Goldschneider et al., 2007; Rehg et al., 2005), content and composition of suspended load and transported bed load material (Bouwer, 2002; Holländer et al., 2005), and the shear forces (Hubbs, 2006b; Schubert, 2006b) scouring out the deposited material on the riverbed (Hubbs, 2006a; Mucha et al., 2006), which in turn are seasonally variable.

Generally, the suspended sediments load carried by the rivers during rainy season is higher than the found during dry season (Dunlop et al., 2008; Göransson et al., 2013); however, in regulated river systems seasonal variations in load does not always follow such a trend (Göransson et al., 2013). Shear forces are also seasonally variable, since these forces are a function of the water level (Hubbs, 2006b). As stated by Regnery et al. (2015), high discharge rates create higher flow velocities and shear stress, which usually result in higher infiltration rates indicating a lower degree of clogging. By contrast, low discharge rates commonly lead to an increase in pore clogging and then to a lower production yield of a RBF system.

4. P7, section 3.1. The reported clogging was is it only a function of the deposition of suspended colloidal matter? What about dissolved organic micro-molecules such as polysaccharides or extracellular polymeric substances?

R/ During the section 3.1 clogging has been attributed to several processes depending on the composition and size of the suspended matter. For instance, P7 line 25 to P8 line 4 reports clogging due to the presence of silt particles, biological growth (e.g. biomass and bacterially produced polysaccharides). The mentioned paragraph is cited below:

"Clogging can be caused by physical, chemical and biological processes, although physical clogging has been found to be the dominant mechanism over the other forms of clogging (Pavelic et al., 2011; Rinck-Pfeiffer et al., 2000). As water flows from the river and through the aquifer to the RBF system, the larger silt particles plug the pore channels to the aquifer in the riverbed and form a less permeable layer together with smaller particles (Grischek and Ray,

2009; Veličković, 2005). Tropical river conditions (temperature and nutrient loads) may be favorable for biological growth onto the riverbed, which might lead to biological clogging (Kim et al., 2010; Platzer and Mauch, 1997; Vandevivere et al., 1995). Rinck-Pfeiffer et al. (2000) reported biological clogging by biomass and bacterially produced polysaccharides in a simulated aquifer storage and recovery wells system, related to the high presence of nutrients. Hoffmann and Gunkel (2011) reported severe clogging mainly induced by biological processes in Lake Tegel reaching a depth of at least 10 cm"

5. Section 2.6 summarizes some interesting findings on micro pollutant removal, the discussion could have been clearer if the mechanisms of removal were also discussed because I don't believe size exclusion played a significant role in removal. Generally micro-pollutants are removed through three possible routes: charge interactions (electrostatic interactions), pollutant-substrate interactions (hydrophobic/hydrophilic interactions) and non-electrostatic interactions (acid-base interactions). Can the authors comment on this?

R/ The mechanisms of water quality improvement in RBF systems are described in section 2.1. In order to strengthen the information regarding to micro-pollutants removal, modifications in sections 1, 2.1 and 2.6 were conducted as follows:

Section 1, page 2, lines 15-18

"In addition, RBF has demonstrated an ability to decrease mutagenic compounds, including naproxen, gemfibrozil and ibuprofen (Hoppe-Jones et al., 2010; Schubert, 2003) and to remove organic and inorganic micro-pollutants, such as sulfamethoxazole and propranolol (Bertelkamp et al., 2014; Hamann et al., 2016; Schmidt et al., 2003). However, it has also been found that specific micro-pollutants such as carbamazepine and EDTA remain mobile, showing a persistent behavior even after 3.6 years of travel time (Hamann et al., 2016). The persistence is mainly driven by the very low reactive and sorptive characteristics of these compounds (Scheytt et al., 2006)."

Section 2.1, page 4, lines 19-22

"Micro-pollutants occur in most surface waters that run through heavily polluted regions or large industrial and agricultural areas. The fate of such substances in RBF systems is mainly determined by sorption mechanisms and biological transformations (Schmidt et al., 2003). During absorption, hydrophobic interactions occur between the aliphatic and aromatic groups of micro-pollutant and the membrane cells of the microorganisms. During adsorption, the negatively charged surfaces of the microorganisms and soil leads to electrostatic interactions of the positively charged micro-pollutants (Luo et al., 2014).

Extensive research in Germany has shown that these compounds may be removed to varying degrees, mainly depending on the properties of each compound (Schmidt et al., 2003). As stated by Schmidt et al. (2004), biodegradation of organic micro-pollutants is a function of the available organic carbon for energy production. The process of energy production is primarily based on redox reactions. The extent of biodegradation of an organic micro-pollutant is dependent on residence time and favorable redox conditions. Therefore, elimination rates of certain micro-pollutants vary depending on local geological and hydrochemical conditions and on organic loads of surface waters and infiltration zones (Schmidt et al., 2004)."

Section 2.6, page 6, lines 23-29

"Hamann et al. (2016) analyzed the fate of 29 micro-pollutant compounds in a RBF system considering a travel time up to 3.6 years, finding complete removal of 14 compounds (2-naphthalene sulfonate, 2,6-NDS, amidotrizoic acid, AMPA, aniline, bezafibrate, diclofenac, ibuprofen, iohexol, iomeprol, iopromide, ioxitalamic acid, metoprolol and sulfamethoxazol) due to retardation and degradation processes as supported from numerical modeling. In addition, some compounds were partially removed (triglyme, iopamidol, diglyme, 1,3,5-naphthalene trisulfonate, 1,3,6-naphthalene trisulfonate), with removal efficiencies ranging from approximately 60 to 90%, based on the highest concentrations measured in both the Lek River and observation well (906 m from river, 3.65 years travel time). Only 10 compounds were fully persistent during the subsurface passage in the RBF system (1,4-dioxan, 1,5-naphthalene disulfonate (1,5-NDS), 2-amino-1,5-NDS, 3-amino-1,5-NDS, AOX, carbamazepine, EDTA, MTBE, toluene and triphenylphosphine oxide). The authors do not differentiate between biodegradation and sorption; where adsorption, ion-pair formation and complexation of pollutants to the soil may lead to soil pollution (Bradl, 2004)."

6.  The impressive removal of some micro-pollutants; could it have been due to the microbial degradation/microbial activity?

R/ Although Hamann et al. (2016) did not mention biodegradation processes, Bertelkamp et al. (2014) (page 7, line 1) and Schmidt et al. (2004) (added below in the response to question 7) stated the biodegradation as an important contributor to the removal of some micro-pollutants.

7.  And if it is degradation, what intermediates/metabolites are formed? Did these studies determine the degradation products?

R/ The cited studies did not report any intermediate/metabolites formation. However, an additional reference is included at the end of section 2.6 (page 7, lines 1-7) reporting metabolites formation.

"…compounds listed before) were completely biodegraded. However, compounds such as atrazine and sulfamethoxazole were not removed in a 6-month period. Drewes et al. (2003) examined the fate of selected pharmaceuticals and personal care products during groundwater recharge, stating that the stimulants caffeine, diclofenac, ibuprofen, ketoprofen, naproxen, fenoproxen and gemfibrozil, were efficiently removed. However, the antiepilectics carbamazepine and primidone were not removed at all. Organic iodine was only partially removed. The formation of metabolites may be expected during organic micro-pollutant biodegradation, however, these have not been reported.

Schmidt et al. (2004) studied the fate of anthropogenic organic micro-pollutants comprising aminopolycarboxylates (EDTA, NTA, DTPA), aromatic sulfonates (2-aminonaphthalene-1,5-NDS, 1,3,6-naphthalene trisulfonate, 1,5-NDS, 1- naphthalene sulfonate, and 2-naphthalene sulfonate), pharmaceutical compounds (diclofenac, carbamazepine, bezafibrate and sulfamethoxazole), iodinated x-ray contrast media (iomeprol, amidotrizoic acid and iopamidol) and MTBE. Schmidt et al. (2004) found that sulfamethoxazole was primarily removed (20% removal efficiency) under anaerobic conditions (anaerobic aquifer), while only slightly reduced in the RBF system under aerobic conditions. The reduction in EDTA concentrations under aerobic conditions was higher than the achieved under denitrifying and anaerobic redox conditions. In addition, the EDTA concentrations in the filtrated water was higher than the

measured in the surface water, concluding that the DTPA was partially biodegraded leading to the formation of EDTA as metabolite (Schmidt et al., 2004)."